# Synthesis, Anticancer Activity and Molecular Docking Studies of Novel *N*-Mannich Bases of 1,3,4-Oxadiazole Based on 4,6-Dimethylpyridine Scaffold

**DOI:** 10.3390/ijms231911173

**Published:** 2022-09-22

**Authors:** Małgorzata Strzelecka, Teresa Glomb, Małgorzata Drąg-Zalesińska, Julita Kulbacka, Anna Szewczyk, Jolanta Saczko, Paulina Kasperkiewicz-Wasilewska, Nina Rembiałkowska, Kamil Wojtkowiak, Aneta Jezierska, Piotr Świątek

**Affiliations:** 1Department of Medicinal Chemistry, Faculty of Pharmacy, Wroclaw Medical University, Borowska 211, 50-556 Wroclaw, Poland; 2Division of Histology and Embryology, Department of Human Morphology and Embryology, Faculty of Medicine, Wroclaw Medical University, Chałubińskiego 6a, 50-368 Wroclaw, Poland; 3Department of Molecular and Cellular Biology, Faculty of Pharmacy, Wroclaw Medical University, Borowska 211, 50-556 Wroclaw, Poland; 4Department of Animal Developmental Biology, Institute of Experimental Biology, University of Wroclaw, Sienkiewicza 21, 50-335 Wroclaw, Poland; 5Department of Bioorganic Chemistry, Faculty of Chemistry, Wroclaw University of Science and Technology, Wybrzeże Wyspiańskiego 27, 50-370 Wroclaw, Poland; 6Faculty of Chemistry, University of Wroclaw, F. Joliot-Curie 14, 50-383 Wroclaw, Poland

**Keywords:** dimethylpyridine, 1,3,4-oxadiazole, *N*-Mannich base, anticancer activity, cytotoxicity, molecular docking, non-covalent interactions

## Abstract

Cancer is one of the greatest challenges in modern medicine today. Difficult and long-term treatment, the many side effects of the drugs used and the growing resistance to treatment of neoplastic cells necessitate new approaches to therapy. A very promising targeted therapy is based on direct impact only on cancer cells. As a continuation of our research on new biologically active molecules, we report herein the design, synthesis and anticancer evaluation of a new series of *N*-Mannich-base-type hybrid compounds containing morfoline or different substituted piperazines moieties, a 1,3,4-oxadiazole ring and a 4,6-dimethylpyridine core. All compounds were tested for their potential cytotoxicity against five human cancer cell lines, A375, C32, SNB-19, MCF-7/WT and MCF-7/DX. Two of the active *N*-Mannich bases (compounds **5** and **6**) were further evaluated for growth inhibition effects in melanoma (A375 and C32), and normal (HaCaT) cell lines using clonogenic assay and a population doubling time test. The apoptosis was determined with the neutral version of comet assay. The confocal microscopy method enabled the visualization of F-actin reorganization. The obtained results demonstrated that compounds **5** and **6** have cytotoxic and proapoptotic effects on melanoma cells and are capable of inducing F-actin depolarization in a dose-dependent manner. Moreover, computational chemistry approaches, molecular docking and electrostatic potential were employed to study non-covalent interactions of the investigated compounds with four receptors. It was found that all the examined molecules exhibit a similar binding affinity with respect to the chosen reference drugs.

## 1. Introduction

Nowadays, the treatment of cancer is an enormous challenge for medicine. In terms of mortality, according to the World Health Organization (WHO), neoplastic disease is the leading cause of death around the world. The most common cancers in 2020 were breast, lung, colon and rectum. High morbidity and mortality result mainly from unhealthy lifestyle, alcohol and tobacco use or lack of physical activity [1]. Due to the diverse pathogenesis of cancer and the multidirectional nature of mutations in the genetic material, treatment is a very long and complicated process. The ability of neoplastic cells to avoid the apoptosis process, induce angiogenesis, stimulate proliferative factors and be insensitive to growth inhibitory signals leads to uncontrolled cell proliferation in the body [2]. In addition, the side effects of anticancer drugs mainly affecting healthy, rapidly dividing cells, e.g., hair loss, neutropenia, nausea and vomiting, are also a limitation in the treatment process [3]. Unfortunately, more and more cancer cells develop multi-drug resistance (MDR) to chemotherapeutic agents, which makes pharmacotherapy much more challenging. The mechanisms of MDR in cancer cells may be due to increased drug efflux, accelerated drug metabolism or various genetic factors [4,5].

Scientists around the world are working on new methods of fighting cancer, which would allow for increased effectiveness of treatment and minimization of undesirable effects of the therapy. Additionally, an increasingly accurate understanding of the molecular pathogenesis of carcinogenesis allows for the implementation of targeted therapy that focuses only on tumor cells and reduces side effects on healthy tissues [6]. A very popular molecular target of new structures is a group of receptors with tyrosine kinase activity. They include, among others, epidermal growth factor (EGFR, HER2) or mesenchymal–epithelial transition factor (c-MET) receptors. An overexpression of the activity of these receptors in the body results in excessive cell proliferation, their increased survival, and the progression of metastases. Additionally, the overtime activity of these receptors may contribute to the increased cell resistance to the treatment [7,8,9]. Another target might be tropomyosin receptor kinase A (TrkA)—overexpression of which leads to the tumor progression and invasions [10]. Molecules that directly and selectively inhibit these types of receptors have a great potential for their use in anticancer therapy. Moreover, there are new approaches to cancer treatment, including immunotherapy, stem cell or gene therapy [11,12].

Multi-component reactions (MCRs) constitute a major part in the present-day organic synthesis in the field of drug design. The Mannich reaction, also named as aminomethylation or aminoalkylation reactions, is a three-component condensation between structurally diverse substrates containing at least one active hydrogen atom, an amine reagent (primary or secondary amines), and an aldehyde component [13]. Mannich bases are known to play a vital role in the development of synthetic pharmaceutical chemistry. By the introduction of a polar functional group, aminomethylation increases the hydrophilic properties of drugs and improves their distribution in the human body. The Mannich reaction can also enhance the lipophilic properties of a drug by selection of the appropriate amine reagent [14]. Studies in the literature revealed that Mannich bases derived from various heterocycles exhibit several biological activities, such as antioxidant [15], analgesic and anti-inflammatory [16,17,18], antimicrobial [19,20,21] and anticonvulsant [22,23] activities. In addition, there is a growing interest in the anticancer activity of Mannich bases. Several classes of NH-azoles have been aminomethylated with a view to synthesize cytotoxic compounds against human cancers, such as lung, gastric, liver, breast, ovarian, prostate and colon cancers [24,25,26,27,28].

On the other hand, 1,3,4-oxadiazoles are of great importance due to their biological activity as well as synthetic applications in medicinal chemistry [29,30,31,32,33]. It is worth mentioning the excellent anticancer activity of 1,3,4-oxadiazole derivatives demonstrated in several studies, both in in vitro and in vivo models. 1,3,4-Oxadiazoles can exert the antitumor activity through multiple mechanisms, such as targeting epidermal growth factor receptors (EGFR, HER2) [34,35,36,37], mesenchymal–epithelial transition factor receptor (c-MET) [38], focal-adhesion kinase (FAK) [28], histone deacetylases (HDAC) [39], telomerase [26,40], thymidylate synthase (TS) [41], tubulin [42] or the DNA structure [27]. 1,3,4-Oxadiazole rings are used as bioisosteres for carbonyl-containing compounds, offering increased water solubility and improved metabolic stability [32,43,44]. Li et al. replaced the amide bond of the scaffold in imatinib, a tyrosine kinase inhibitor used to treat a number of leukemias, to form 1,3,4-oxadiazole analogs of imatinib [45]. This modification enhanced the inhibitory activity against the human leukemia stem-like cell line (KG1a), and the potencies of compound **I** (Figure 1) being over 30 times more remarkable than that of imatinib.

The combination of different pharmacophores in the same unit is an attractive approach to discover novel potent drugs, due to the possible synergistic effect. Several reports of Mannich bases of 1,3,4-oxadiazole rings as cytotoxic agents are available in the recent literature. Anticancer screening studies for a series of *N*-Mannich bases of 5-(quinolin-2-yl)-1,3,4-oxadiazole-2(3*H*)-thione showed that compounds **II** (Figure 1) displayed broad-spectrum antitumor activity against a panel consisting of human hepatoma (HepG2), gastric (SGC-7901) and breast (MCF-7) cancer cell lines using the MTT (3-[4,5-dimethylthiazol-2-yl]-2,5-diphenyltetrazolium bromide) method, and were more potent (2.5- or even 27-fold) compared to that of 5-fluorouracil, widely used in the treatment of cancer [26]. Moreover, the tested compounds **II** exhibited a potent telomerase inhibitory potency with IC_50_ ranging from 0.8 to 0.9 µM. Another 1,3,4-oxadiazole-2(3*H*)-thione derivative containing a phenylpiperazine skeleton **III** (Figure 1) exhibited a stronger cytotoxic effect on hepatoma cancer cells (HepG2) with 2.3-fold higher activity than the reference 5-fluorouracil. Additional studies for focal-adhesion kinase inhibition showed remarkable in vitro inhibitory activity of compound **III** (IC_50_ = 0.78 µM) supported by molecular docking of this compound into active site of FAK [28].

In our recently published work, we demonstrated that 1,3,4-oxadiazole derivatives of 4,6-dimethylpyridine **IV** (Figure 1) containing differently substituted *N*-acyl hydrazone moieties exhibited potent anticancer activity against a panel consisting of human lung (A549), breast (MCF-7) and colon (LoVo) and its drug-resistant subline LoVo/Dx cancer cell lines [46]. Encouraged by those promising results, we decided to modify the structure of the above-mentioned derivatives by replacing the Schiff base-type pharmacophore at position 4 of 1,3,4-oxadiazole with secondary amines linked to the heterocycle by a methylene bridge (Figure 1). The introduction of this aminomethyl function, which resulted in a new series of *N*-Mannich bases, was inspired by the leading research described in the previously cited literature. By such modification, we wanted to determine the impact of the aryl/heteroaryl/alkylpiperazine or morpholine residues on the cytotoxic activity of the compounds and selectivity towards cancer cell lines.

The new compounds were examined for their potential cytotoxicity against selected human cancer cell lines: melanotic (A375) and amelanotic (C32) melanoma, glioblastoma (SNB-19), breast adenocarcinoma (MCF-7/WT) and drug-resistant breast adenocarcinoma (MCF-7/DX); and, additionally, normal cells—human keratinocytes (HaCaT)—were included in the study. Two of the compounds (**5** and **6**) that displayed promising cytotoxic activity in preliminary study were further evaluated for growth inhibition effects in melanoma (A375 and C32) and normal (HaCaT) cell lines using clonogenic assay and a population doubling time test. The apoptosis was determined with the neutral version of comet assay. The confocal microscopy method enabled the visualization of F-actin reorganization. Our experimental findings were supported by computational chemistry approaches: molecular docking and electronic structure study on the basis of electrostatic potential maps (EPMs).

## 2. Results and Discussion

### 2.1. Chemistry

The synthesis of *N*-(2-hydrazinyl-2-oxoethyl)-4,6-dimethyl-2-sulfanylpyridine-3-carboxamide **1** was performed according to the protocols published previously [47]. Figure 1 presents the synthesis of compounds that have not been described in the literature yet. The spectroscopic properties of all newly obtained derivatives were in good agreement with their predicted structures and are summarized in the experimental section. The formation of final *N*-Mannich bases **3**–**12** was achieved via a convenient and efficient one-step reaction of compound **2** with appropriate secondary amines (piperazine derivatives or morpholine) and formaldehyde in ethanol. The structures of the various synthesized compounds were determined based on spectral data analysis, such as FT-IR, ^1^H NMR, ^13^C NMR and MS.

The FT-IR spectra of compounds **3**–**12** showed peaks around 1650–1660 cm^−1^ due to carbonyl function derived from the amide structure. Additionally, the IR spectra exhibited, in the 3285–3155 cm^−1^ range, the NH weak band of the CONH functions.

The distinctive peak in the ^1^H NMR spectrum near δ 5.00 ppm and the signal at around δ 70.00 ppm in the ^13^C NMR spectrum clearly indicate the formation of the methylene linker characteristic for Mannich bases. Additionally, in the ^1^H NMR spectra of the final compounds, the signals of the piperazine or morpholine protons, in the form of two four-proton multiplets in the range of 2.38–3.73 ppm, were recorded. All NMR spectra are presented in Appendix A.

The HRMS (ESI-MS) of **3**–**12** showed the characteristic corresponding peaks to their molecular formula.

### 2.2. Biological Tests

#### 2.2.1. MTT Cell Viability Assay

The cell viability assay in cytotoxic evaluation is a major step in analyzing the cellular response to toxic compounds and plays a crucial role in determining the cell survival rate and assessment of metabolic activity. The preliminary cytotoxicity study of *N*-Mannich bases **3**–**12** was carried out on five human cancer cell lines: melanotic (A375) and amelanotic (C32) melanoma, glioblastoma (SNB-19), and sensitive (MCF-7/WT) and doxorubicin-resistant (MCF-7/DX) breast adenocarcinoma, using the MTT colorimetric method. The obtained results, shown in Figure 2, demonstrated the highest anticancer potential of compounds **5** and **6**, containing 3,4-dichloro- and 3-trifluorophenylpiperazine moieties, respectively, and these two were selected for the more detailed study. There were selected skin cancer cell lines (A375 and C32), and, additionally, normal cells—human keratinocytes (HaCaT)—were included in the study. The response of cells to incubation with *N*-Mannich bases varied in different cell lines (Figure 3). Both skin cancers and normal cells were highly sensitive to the growth inhibitory activity of compound **5** at a concentration of 100 µM. In the case of compound **6**, melanomas A375 and C32 were more affected at lower concentrations in comparison to keratinocytes. It is worth noting that the A375 cell line was more sensitive to both *N*-Mannich bases than the C32 cell line. The most significant cytotoxic effect was observed for compound **5** against A375 cells (IC_50_ = 80.79 µM) (Table 1). This indicates the cytotoxicity of compounds at low concentrations and short incubation time, which was confirmed by the population doubling time test.

#### 2.2.2. Clonogenic Assay

The colony formation assay was used to determine the long-term cytotoxic effect on the growth of cancer cells. In Figure 4a,b are shown the results obtained from the clonogenic assay after exposure of melanoma cells and keratinocytes to compounds **5** and **6**. It was noted that compound **5** significantly inhibited colony formation in A375 cells at two concentrations, 50 and 100 µM. However, the highest cytotoxic effect of compound **5** was observed among human keratinocytes. In the case of the C32 cell line, the results were comparable to the level of control cells. Compound **6** reduced the colony growth of all cell lines in a dose-dependent manner, but at a higher concentration than compound **5**. Figure 4c shows cells plated for clonogenic assay, with characteristic stained colonies.

#### 2.2.3. Population Doubling Time

The results of the doubling time are summarized in Figure 5. Cells were seeded with a plating density of 3000 viable cells. Figure 5 shows growth curves from independent experiments of subcultured cells. The data are presented as population doubling (PD) versus the time. PD was calculated as log2 (number of viable cells/number of plated cells). The growth of the curves was observed for both compounds, with low concentrations showing a logarithmic increase. However, high concentrations showed a loss of cell population all days after seeding, followed by a logarithmic decrease. Untreated cells revealed a logarithmic increase in growth one day after cultivation.

#### 2.2.4. Cell Death Detection by Comet Assay

Detection of DNA damage and cell death was investigated by means of the neutral comet assay, where we could distinguish between three types of comets showing late and early apoptosis and not-affected cells. In Figure 6a–c are shown the results obtained from the 24 h exposure to compounds **5** and **6**. We could observe the highest percentage of early apoptotic cells for compound **6** (100 µM) in A375 cells, and late apoptosis was detected for higher concentrations (300 µM). C32 cells were less sensitive, and the percentage of DNA damage was lower than in A375 cells. The obtained results are also confirmed by the olive tail moment (OTM) calculations (Figure 6b), which correspond to the product of the tail length and the fraction of total DNA in the tail. The longest tail was observed in the case of A375 cells exposed to 300 µM of compound **6**. C32 cells revealed the longest tail after the exposure to 100 µM of compound **5** and to 200 µM of compound **6**. Both cell lines were less sensitive to compound **5**, but 100 µM concentration stimulated cells to the early apoptotic state.

#### 2.2.5. Fluorescent Staining of Actin Filaments

The visualization of the F-actin organization in normal and cancer cells is presented in Figure 7. The 24 h exposure to the tested compounds demonstrated the most significant changes in the cytoskeleton organization in all cell lines, after the treatment with compound **5** at 100 µM concentration and with compound **6** at 200 and 300 µM concentration. The compound **6** in 100 µM concentration did not affected normal keratinocytes but significantly damaged melanoma cells, causing cells’ shrinkage and reduced cells’ number. Normal keratinocytes (HaCaT) were also not sensitive to compound **5** in 25 and 50 µM concentrations.

### 2.3. Molecular Docking Studies

Docking studies were performed to assess the binding affinity of compounds **3**–**12** and the reference drugs to the selected receptors: cMet (PDB code: 3RHK) [52], EGFR (PDB code: 5GTY) [53], HER2 (PDB code: 7JXH) [54] and hTrkA (PDB code: 6PL2) [55]. The four chosen receptors are well known for their importance in cancer progression and metastasis. The ligands denoted as M97 [52], 816 [53], VOY [54] and OOM [55] were re-docked to the receptors. The positions of the co-crystallized ligands with the lowest binding affinity values were selected and presented in Figure 8. Careful inspection of Figure 8 shows that the docking parameters were chosen appropriately, because in the case of the M97, 816, VOY and OOM ligands, the root-mean-square deviation (RMSD) values are relatively low and they are equal to: 0.832, 1.812, 2.006 and 1.200 Å, respectively.

After the validation of the docking protocol, compounds **3**–**12** were docked and their binding affinity was estimated—their performance was compared to the binding affinities of known inhibitors of the cMet, EGFR, HER2 and hTrkA receptors, namely: Erlotinib, Neratinib and Tepotinib (see Table 2).

Most of the compounds from the set of **3**–**12**, with regards to each of the receptors, obtained a lower binding affinity score than one of the reference drugs (Erlotinib)—the exceptions were only complexes with **8**-EGFR and **9**-HER2. The most interesting compounds, when the binding affinity to the chosen receptors is taken into consideration, were compounds **7** and **11**. Compound **7** had the best score of binding to the receptors EGFR and HER2, where it values were equal to −12.9 and −13.6 kcal/mol, respectively. In turn, compound **11** was bound in the most pronounced way by the cMet (−12.6 kcal/mol) and hTrkA (−14.5 kcal/mol) receptors. The binding modes to the four receptors with compounds **7** and **11** are presented in Figure 9.

It is visible that the flexible binding sites, depending on the receptor, varied in size. It is especially noticeable in the binding pocket of cMet, which had only **7** flexible amino acid residues. In the case of the remaining three receptors, the flexible parts consisted of 19 (for EGFR), 18 (for HER2) and 14 (for hTrkA) residues, respectively. In order to perform a more in-depth analysis of the binding modes of the abovementioned structures, 2D diagrams of the ligand–receptor interactions were prepared (see Figure 10).

An inspection of the presented diagrams shows that the **11**-cMet complex interactions are stabilized mainly by the presence of hydrogen bonds, in which the sulfur atom of the thiol group attached to pyridine and the carbonyl oxygen acts as an electron density donor to the Met1269 of the binding pocket. Other important interactions, such as Van der Waals with Tyr1159; π-alkyl with Ile1084, Phe1089, Leu1157 and Leu1140; and π-σ with Phe1223 are also present and stabilize the complex. For the complex of **11**-hTrkA, the interactions present in the binding pocket differ mainly by the contribution of many the Van der Waals contacts of Phe646, Phe669, Leu657, Ile572, Glu560 and Tyr591 to the stabilization of the examined compound. The binding of the ligand to the receptor is also stabilized by the π-σ interactions of Val524 with the aromatic ring of pyridine and the hydrogen bond formed by His648 (which acts as a proton donor) and the oxygen from the 1,3-benzodioxole moiety. As was the case with the former, π-alkyl interactions are also present (for more details, see Figure 10). Moreover, the binding affinity for this complex is equal to -14.5 kcal/mol. It is the lowest value among the studied, synthesized **3**–**12** and reference compounds (as it is shown in Table 2). In the case of the 7-EGFR complex, the most important contributions to the binding affinity come from the presence of the hydrogen bond between the Asp668 and the hydrogen atom from the amide group of the compound **7** as well as from the π-σ interactions of **7** with the Leu718 and Met780 residues. A plethora of π-alkyl interactions with various residues is present as well. A totally different mode of binding exists in the case of the 7-HER2 complex—here, four different hydrogen bonds, one π-sulfur and two π-alkyl interactions are formed between the ligand and the binding site of the receptor. Two of the hydrogen bonds can be classified as weak hydrogen bonds, where the carbon atom of the Asp863 and the carbon atom from the methyl group attached to pyridine act as proton donors. Another two, conventional hydrogen bonds were formed between **7** and the residues Thr798 and Lys753, where amino acids act as proton donors. Additionally, contributions from the π-sulfur and π-alkyl interactions were noticed as well.

As a supplement to our discussion of the ligand–receptor interactions, the Molecular Electrostatic Potential (MEP) maps of compounds **7** and **11** as well as Neratinib and Tepotinib were prepared (see Figure 11).

From the perspective of the detailed description of the possible interactions of the ligands with the binding pocket, one must also take into consideration the possible anisotropy of the charge density distribution—which is not taken into account by modern docking software [56]. The arrows depicted in Figure 11 point to the MEP extrema relevant to the analysis of the interactions involving the ligand. For compound **6**, only negative extrema are presented—in fact, there is no σ-hole or π-hole at the CF_3_ substituent and at the center of the benzene ring, respectively. Due to that, CF_3_ can act only as an electron density—which is a distinguishing feature between **5** and **6**, because in the case of compound **5**, we can observe two σ-holes (of magnitude 0.41 and 1 kcal/mol) on chlorine atoms attached to the benzene ring (in this manner, compound **5** can form two halogen bonds). With regards to compound **7**, it can be noted that four different extrema exist. One can observe the presence of a π-hole on the nitrogen atom of the nitro group, a σ-hole on the nitrogen from the 1,3,4-oxadiazole ring and a π-hole on the sulfur atom of the thiol group attached to the pyridine ring with 2.69, −0.84 and −0.54 kcal/mol MEP values, respectively. On the sulfur atom of the thiol group, at the opposite site, there is also present another extremum with a −17.71 kcal/mol value of the MEP. These numbers indicate that the first three abovementioned atoms can act as acceptors of the electron density and take a part in the σ- and π-hole interactions. Interestingly, on the sulfur atom of the thiol group, two distinct extrema were spotted; thus, this atom could act as a Lewis-acid as well as Lewis-base center in the intermolecular interactions. The analysis of the MEP corresponding to compound **11** is somewhat similar. In fact, the one important difference is the ability of the terminal phenyl ring to form π-hole interactions—on the basis of MEP analysis one can suppose that the quadrupole moment of the phenyl in compound **7** is higher compared to the same aromatic framework in compound **11**, due to presence of the NO_2_ group (which is able to withdraw electrons). In the case of Neratinib, two spots were noticed—one σ-hole on the chlorine atom (2.31 kcal/mol) and the π-hole at the pyridine (−3.32 kcal/mol). It is noteworthy that a similarly positioned π-hole (5.19 kcal/mol) is present in the structure of Tepotinib. In fact, both Neratinib and Tepotinib possess a highly electron-withdrawing –CN group strongly affecting the charge distribution of the molecules.

## 3. Materials and Methods

### 3.1. Chemistry

#### 3.1.1. Instruments and Chemicals

All solvents, reagents and chemicals used during the experiments described in this paper were delivered by commercial suppliers (Alchem, Wrocław, Poland; Chemat, Gdańsk, Poland; Archem, Łany, Poland) and were used without further purification. Any dry solvents were received due to standard procedures. Reaction progress was monitored by the Thin-Layer Chromatography (TLC) technique, on TLC plates made of 60–254 silica gel, and was visualized by UV light at 254/366 nm. Melting points of final compounds were determined on an Electrothermal Mel-Temp 1101D apparatus (Cole-Parmer, Vernon Hills, IL, USA) using the open capillary method, no correction needed. ^1^H NMR (300 MHz) and ^13^C NMR (75 MHz) spectra were recorder using a Bruker 300 MHz NMR spectrometer (Bruker Analytische Messtechnik GmbH, Rheinstetten, Germany) in DMSO-*d_6_*, with tetramethylsilane (TMS) as an internal reference. Chemical shifts (δ) were reported in ppm. In order to record and read spectra, the TopSpin 3.6.2. (Bruker Daltonik, GmbH, Bremen, Germany) program was used. FT-IR spectra were measured on a Nicolet iS50 FT-IR Spectrometer (Thermo Fisher Scientific, Waltham, MA, USA). Frequencies were reported in cm^−1^. All samples were solid, and spectra were read by OMNIC Spectra 2.0 (Thermo Fisher Scientific, Waltham, MA, USA). Mass spectra (MS) were recorded using the Bruker Daltonics Compact ESI-Mass Spectrometer (Bruker Daltonik, GmbH, Bremen, Germany), operating in the positive ion mode with methanol as a solvent. Theoretical monoisotopic masses of ions were calculated (calcd.) using Bruker Compass Data Analysis 4.2 software (Bruker Daltonik GmbH, Bremen, Germany).

#### 3.1.2. Preparation and Experimental Properties of Compounds **3**–**12**

The synthesis protocols and experimental data for compound **1** and **2** were already reported [46,47].

General Procedure for Preparation of Compounds **3**–**12**

0.16 mL of 36% formaldehyde was added to a solution of 0.001 mol of 4,6-dimethyl-*N*-[(5-sulfanylidene-4,5-dihydro-1,3,4-oxadiazol-2-yl)methyl]-2-sulfanylpyridine-3-carboxamide **2** in 30 mL of ethanol. The obtained mixture was stirred at room temperature for several minutes. Then, 0.001 mole of the appropriate piperazine derivative or morpholine was added to the flask. The resulting mixture was stirred for 4 h at room temperature and then left overnight. The obtained precipitate was filtered off and allowed to dry, then the obtained product was crystallized from ethanol.

4,6-Dimethyl-*N*-{[4-((4-phenylpiperazin-1-yl)methyl)-5-sulfanylidene-4,5-dihydro-1,3,4-oxadiazol-2-yl]methyl}-2-sulfanylpyridine-3-carboxamide **3**

Yield: 52.0%, m.p: 215–218 °C

FT-IR (selected lines, γ_max_, cm^−1^): 3193 (NH), 2926, 2824 (C-H aliph.), 1663 (C=O)

^1^H NMR (300 MHz, DMSO-*d_6_*): δ = 2.07 (s, 3H, CH_3_), 2.26 (s, 3H, CH_3_), 2.81–2.83 (m, 4H, CH_2-piperazine_), 3.09–3.11 (m, 4H, CH_2-piperazine_), 4.44–4.46 (d, 2H, CH_2_, *J = 6 Hz*), 5.00 (s, 2H, CH_2_), 6.48 (s, 1H, H_-pyridine_), 6.73–6.78 (m, 1H, ArH), 6.89–6.91 (m, 2H, ArH), 7.15–7.20 (m, 2H, ArH), 8.84 (t, 1H, NH, *J = 6 Hz*), 13.29 (s, 1H, SH);

^13^C NMR (75 MHz, DMSO-*d_6_*): δ = 18.74, 19.32, 34.67, 48.76, 50.01, 69.96, 115.28, 116.13, 119.52, 129.43, 146.65, 148.52, 151.51, 167.45, 174.32

HRMS (ESI-MS) (*m/z*): calcd. for C_22_H_26_N_6_O_2_S_2_ [M+H]^+^: 471.1631; found: 471.1634

4,6-Dimethyl-*N*-{[4-((4-(2-chloro)phenylpiperazin-1-yl)methyl)-5-sulfanylidene-4,5-dihydro-1,3,4-oxadiazol-2-yl]methyl}-2-sulfanylpyridine-3-carboxamide **4**

Yield: 62.7%, m.p: 208–210 °C

FT-IR (selected lines, γ_max_, cm^−1^): 3193 (NH), 2827 (C-H aliph.), 1655 (C=O)

^1^H NMR (300 MHz, DMSO-*d_6_*): δ = 2.08 (s, 3H, CH_3_), 2.27 (s, 3H, CH_3_), 2.85–2.87 (m, 4H, CH_2-piperazine_), 2.94–2.96 (m, 4H, CH_2-piperazine_), 4.47–4.49 (d, 2H, CH_2_, *J = 6 Hz*), 5.00 (s, 2H, CH_2_), 6.49 (s, 1H, H_-pyridine_), 7.01–7.03 (m, 1H, ArH), 7.14–7.16 (m, 1H, ArH), 7.26–7.28 (m, 1H, ArH), 7.36–7.38 (m, 1H, ArH), 8.86 (t, 1H, NH, *J = 6 Hz*), 13.28 (s, 1H, SH)

^13^C NMR (75 MHz, DMSO-*d_6_*): δ = 18.70, 19.39, 34.63, 50.25, 51.23, 70.28, 115.21, 115.26, 121.58, 124.46, 127.93, 128.52, 130.79, 146.87, 148.51, 151.62, 167.49, 174.59

HRMS (ESI-MS) (*m/z*): calcd. for C_22_H_25_ClN_6_O_2_S_2_ [M+H]^+^: 505.1242; found: 505.1224

4,6-Dimethyl-*N*-{[4-((4-(3,4-dichloro)phenylpiperazin-1-yl)methyl)-5-sulfanylidene-4,5-dihydro-1,3,4-oxadiazol-2-yl]methyl}-2-sulfanylpyridine-3-carboxamide **5**

Yield: 70.3%, m.p: 226–228 °C

FT-IR (selected lines, γ_max_, cm^−1^): 3158 (NH), 2834 (C-H aliph.), 1646 (C=O)

^1^H NMR (300 MHz, DMSO-*d_6_*): δ = 2.07 (s, 3H, CH_3_), 2.26 (s, 3H, CH_3_), 2.79–2.81 (m, 4H, CH_2-piperazine_), 3.15–3.17 (m, 4H, CH_2-piperazine_), 4.45–4.47 (d, 2H, CH_2_, *J = 6 Hz*), 5.00 (s, 2H, CH_2_), 6.48 (s, 1H, H_-pyridine_), 6.89–6.93 (m, 1H, ArH), 7.09–7.10 (m, 1H, ArH), 7.34–7.37 (m, 1H, ArH), 8.81 (t, 1H, NH, *J = 6 Hz*), 13.27 (s, 1H, SH)

^13^C NMR (75 MHz, DMSO-*d_6_*): δ = 18.70, 19.45, 34.72, 48.00, 49.70, 70.02, 115.25, 116.06, 116.82, 120.19, 125.00 130.87, 146.73, 148.50, 151.30, 153.20, 167.56, 174.47

HRMS (ESI-MS) (*m/z*): calcd. for C_22_H_24_Cl_2_N_6_O_2_S_2_ [M+H]^+^: 539.0852; found: 539.0836

4,6-Dimethyl-*N*-{[4-((4-(3-trifluoromethyl)phenylpiperazin-1-yl)methyl)-5-sulfanylidene-4,5-dihydro-1,3,4-oxadiazol-2-yl]methyl}-2-sulfanylpyridine-3-carboxamide **6**

Yield: 59.3%, m.p: 215–218 °C

FT-IR (selected lines, γ_max_, cm^−1^): 3164 (NH), 2835 (C-H aliph.), 1648 (C=O)

^1^H NMR (300 MHz, DMSO-*d_6_*): δ = 2.07 (s, 3H, CH_3_), 2.26 (s, 3H, CH_3_), 2.82–2.84 (m, 4H, CH_2-piperazine_), 3.19–3.21 (m, 4H, CH_2-piperazine_), 4.45-4.47 (d, 2H, CH_2_, *J = 6 Hz*), 5.01 (s, 2H, CH_2_), 6.48 (s, 1H, H_-pyridine_), 7.04–7.06 (m, 1H, ArH), 7.13–7.21 (m, 1H, ArH), 7.36–7.42 (m, 1H, ArH), 8.83 (t, 1H, NH, *J =* 6 *Hz*), 13.27 (s, 1H, SH)

^13^C NMR (75 MHz, DMSO-*d_6_*): δ = 18.70, 19.37, 34.77, 48.10, 49.84, 70.28, 111.61, 115.21, 115.26, 119.51, 130.39, 136.66, 146.14, 148.57, 151.49, 167.78, 174.84

HRMS (ESI-MS) (*m/z*): calcd. for C_23_H_25_F_3_N_6_O_2_S_2_ [M+H]^+^: 539.1505; found: 539.1534

4,6-Dimethyl-*N*-{[4-((4-(4-nitro)phenylpiperazin-1-yl)methyl)-5-sulfanylidene-4,5-dihydro-1,3,4-oxadiazol-2-yl]methyl}-2-sulfanylpyridine-3-carboxamide **7**

Yield: 67.3%, m.p: 221–223 °C

FT-IR (selected lines, γ_max_, cm^−1^): 3157 (NH), 2834 (C-H aliph.), 1647 (C=O)

^1^H NMR (300 MHz, DMSO-*d_6_*): δ = 2.04 (s, 3H, CH_3_), 2.26 (s, 3H, CH_3_), 2.80–2.82 (m, 4H, CH_2-piperazine_), 3.44–3.46 (m, 4H, CH_2-piperazine_), 4.42–4.44 (d, 2H, CH_2_, *J = 6 Hz*), 5.02 (s, 2H, CH_2_), 6.46 (s, 1H, H_-pyridine_), 6.99–7.02 (d, 2H, ArH), 8.00–8.03 (d, 2H, ArH), 8.82 (t, 1H, NH, *J =* 6 *Hz*), 13.28 (s, 1H, SH)

^13^C NMR (75 MHz, DMSO-*d_6_*): δ= 18.68, 19.35, 34.50, 46.76, 49.65, 69.82, 106.70, 113.18, 115.19, 120.44, 126.17, 126.41, 127.67, 146.65, 148.59, 150.21, 167.46, 175.44

HRMS (ESI-MS) (*m/z*): calcd. for C_22_H_25_N_7_O_4_S_2_ [M+H]^+^: 516.1482; found: 516.1465

4,6-Dimethyl-*N*-{[4-((4-(2-methoxy)phenylpiperazin-1-yl)methyl)-5-sulfanylidene-4,5-dihydro-1,3,4-oxadiazol-2-yl]methyl}-2-sulfanylpyridine-3-carboxamide **8**

Yield: 64.0%, m.p: 207–210 °C

FT-IR (selected lines, γ_max_, cm^−1^): 3176 (NH), 2830 (C-H aliph.), 1653 (C=O)

^1^H NMR (300 MHz, DMSO-*d_6_*): δ = 2.09 (s, 3H, CH_3_), 2.27 (s, 3H, CH_3_), 2.81–2.83 (m, 4H, CH_2-piperazine_), 2.91–2.93 (m, 4H, CH_2-piperazine_), 3.73 (s, 3H, OCH_3_), 4.47–4.49 (d, 2H, CH_2_, *J = 6 Hz*), 4.99 (s, 2H, CH_2_), 6.49 (s, 1H, H_-pyridine_), 6.85–6.91 (m, 4H, ArH), 8.85 (t, 1H, NH, *J = 6 Hz*), 13.29 (s, 1H, SH)

^13^C NMR (75 MHz, DMSO-*d_6_*): δ = 18.70, 19.38, 34.64, 50.27, 50.36, 50.44, 55.68, 70.12, 112.18, 115.22, 118.49, 121.20, 123.00, 136.76, 141.49, 146.68, 148.49, 152.39, 152.45, 167.48, 174.41, 178.21

HRMS (ESI-MS) (*m/z*): calcd. for C_23_H_28_N_6_O_3_S_2_ [M+H]^+^: 501.1737; found: 501.1719

4,6-Dimethyl-*N*-{[4-(morpholinyl)methyl)-5-sulfanylidene-4,5-dihydro-1,3,4-oxadiazol-2-yl]methyl}-2-sulfanylpyridine-3-carboxamide **9**

Yield: 79.0%, m.p: 178–181 °C

FT-IR (selected lines, γ_max_, cm^−1^): 3167 (NH), 2859 (C-H aliph.), 1652 (C=O)

^1^H NMR (300 MHz, DMSO-*d_6_*): δ = 2.07 (s, 3H, CH_3_), 2.27 (s, 3H, CH_3_), 2.66–2.68 (m, 4H, CH_2-morpholine_), 3.53–3.55 (m, 4H, CH_2-morpholine_), 4.44-4.46 (d, 2H, CH_2_, *J = 6 Hz*), 4.92 (s, 2H, CH_2_), 6.49 (s, 1H, H_-pyridine_), 8.84 (t, 1H, NH, *J = 6 Hz*), 13.23 (s, 1H, SH)

^13^C NMR (75 MHz, DMSO-*d_6_*): δ = 18.69, 19.37, 34.59, 43.20, 49.04, 50.35, 63.47, 66.45, 70.03, 115.22, 136.90, 146.64, 148.82, 167.54, 174.44

HRMS (ESI-MS) (*m/z*): calcd. for C_16_H_21_N_5_O_3_S_2_ [M+H]^+^: 396.1159; found: 396.1165

4,6-Dimethyl-*N*-{[4-((4-(pyrimidin-2-yl)piperazin-1-yl)methyl)-5-sulfanylidene-4,5-dihydro-1,3,4-oxadiazol-2-yl]methyl}-2-sulfanylpyridine-3-carboxamide **10**

Yield: 63.8%, m.p: 234–236 °C

FT-IR (selected lines, γ_max_, cm^−1^): 3285 (NH), 2971, 2926, 2830 (C-H aliph.), 1663 (C=O)

^1^H NMR (300 MHz, DMSO-*d_6_*): δ = 2.04 (s, 3H, CH_3_), 2.26 (s, 3H, CH_3_), 2.71–2.73 (m, 4H, CH_2-piperazine_), 3.71-3.73 (m, 4H, CH_2-piperazine_), 4.42–4.44 (d, 2H, CH_2_, *J = 6 Hz*), 5.00 (s, 2H, CH_2_), 6.47 (s, 1H, H_-pyridine_), 6.57–6.61 (m, 1H, ArH), 8.31–8.33 (m, 2H, ArH), 8.80 (t, 1H, NH, *J = 6 Hz*), 13.27 (s, 1H, SH)

^13^C NMR (75 MHz, DMSO-*d_6_*): δ = 18.64, 19.38, 34.58, 43.53, 49.88, 70.06, 110.67, 115.17, 136.73, 146.56, 148.48, 158.38, 161.51, 167.44, 174.38

HRMS (ESI-MS) (*m/z*): calcd. for C_20_H_24_N_8_O_2_S_2_ [M+H]^+^: 473.1536; found: 473.1528

4,6-Dimethyl-*N*-{[4-((4-(1,3-benzodioxol-5-ylmethyl)piperazin-1-yl)methyl)-5-sulfanylidene-4,5-dihydro-1,3,4-oxadiazol-2-yl]methyl}-2-sulfanylpyridine-3-carboxamide **11**

Yield: 35.8%, m.p: 210–212 °C

FT-IR (selected lines, γ_max_, cm^−1^): 3155 (NH), 2911, 2839 (C-H aliph.), 1646 (C=O)

^1^H NMR (300 MHz, DMSO-*d_6_*): δ = 2.07 (s, 3H, CH_3_), 2.27 (s, 3H, CH_3_), 2.34–2.36 (m, 4H, CH_2-piperazine_), 2.67–2.69 (m, 4H, CH_2-piperazine_), 4.44–4.46 (d, 2H, CH_2_, *J = 6 Hz*), 4.92 (s, 2H, CH_2_), 5.96 (s, 2H, CH_2-benzodioxole_), 6.48 (s, 1H, H_-pyridine_), 6.69-6.72 (m, 1H, ArH), 6.80–6.82 (m, 2H, ArH), 8.80 (t, 1H, NH, *J = 6 Hz*), 13.27 (s, 1H, SH)

^13^C NMR (75 MHz, DMSO-*d_6_*): δ = 18.70, 19.40, 34.60, 49.82, 52.56, 61.88, 70.38, 101.24, 108.29, 109.54, 115.22, 122.52, 130.49, 136.80, 146.68, 147.65, 148.47, 167.45, 174.42

HRMS (ESI-MS) (*m/z*): calcd. for C_24_H_28_N_6_O_4_S_2_ [M+H]^+^: 529.1686; found: 529.1678

4,6-Dimethyl-*N*-{[4-((4-hexylpiperazin-1-yl)methyl)-5-sulfanylidene-4,5-dihydro-1,3,4-oxadiazol-2-yl]methyl}-2-sulfanylpyridine-3-carboxamide **12**

Yield: 31.3%, m.p: 279–283 °C

FT-IR (selected lines, γ_max_, cm^−1^): 3179 (NH), 2928, 2857 (C-H aliph.), 1642 (C=O)

^1^H NMR (300 MHz, DMSO-*d_6_*): δ = 0.83 (t, 3H, CH_3_, *J = 6 Hz*), 1.20–1.24 (m, 6H, CH_2_), 1.41–1.43 (m, 2H, CH_2_), 2.08 (s, 3H, CH_3_), 2.15–2.17 (m, 2H, CH_2_), 2.27 (s, 3H, CH_3_), 2.38–2.40 (m, 4H, CH_2-piperazine_), 2.73-2.75 (m, 4H, CH_2-piperazine_), 4.45-4.47 (d, 2H, CH_2_, *J = 6 Hz*), 4.93 (s, 2H, CH_2_), 6.49 (s, 1H, H_-pyridine_), 8.86 (t, 1H, NH, *J = 6 Hz*), 13.29 (s, 1H, SH)

^13^C NMR (75 MHz, DMSO-*d_6_*): δ = 14.35, 18.70, 19.41, 22.46, 25.06, 26.76, 31.51, 34.64, 49.09, 52.49, 57.87, 69.79, 115.22, 115.31, 136.66, 146.66, 148.44, 167.54, 174.41

HRMS (ESI-MS) (*m/z*): calcd. for C_22_H_34_N_6_O_2_S_2_ [M+H]^+^: 479.2257; found: 479.2250

### 3.2. Biological Section

#### 3.2.1. Cell Lines

The following cell lines were used in the study: human melanotic melanoma cell line A375 (CRL-1619™); human amelanotic melanoma cell line C32 (CRL-1585™); human glioblastoma SNB-19 (CRL-2219™); two breast adenocarcinoma cell lines: sensitive MCF-7/WT and resistant MCF-7/DX; and immortalized human keratinocyte from histologically normal skin HaCaT, purchased from the American Type Culture Collection (ATCC^®^). Breast cancer cell lines were a kind gift from the Department of Experimental and Clinical Radiobiology, Center of Oncology (Gliwice, Poland). Cells were cultured as a monolayer in Dulbecco’s Modified Eagle’s Medium (DMEM, Sigma-Aldrich, St. Louis, MO, USA). The medium was supplemented with 10% fetal bovine serum (FBS, Sigma-Aldrich, St. Louis, MO, USA) and 1% of antibiotic (streptomycin/penicillin, Sigma-Aldrich, St. Louis, MO, USA). The cells were incubated at 37 °C in a humidified atmosphere containing 5% CO_2_. The cell medium was changed 2–3 times per week. For the experimental protocols, cells were washed with phosphate-buffered saline (PBS) and removed by trypsinization (0.025% trypsin and 0.02% EDTA; Sigma-Aldrich, St. Louis, MO, USA).

#### 3.2.2. MTT Cell Viability Assay

The evaluation of a potential cytotoxic action of the compounds was performed in monolayer culture on human cancer cell lines (A375, C32, SNB-19, MCF-7/WT and MCF-7/DX) and a normal cell line: HaCaT. Stock solutions were prepared in DMSO (dimethyl sulfoxide, Sigma Aldrich, St. Louis, MO, USA); and compound dilutions were performed in Dulbecco’s Modified Eagle’s Medium supplemented with 10% FBS (EURx, Gdansk, Poland), where DMSO concentration did not exceed more than 1% in the sample. Compounds were tested in the 25–300 µM concentration range. The cells were seeded in 96-well flat-bottom plates at a density of 3 × 10^4^ cells/well, and cells were incubated for 24 h in a cell culture incubator for the cells to stick to the plate. After the incubation, the culture supernatants were removed, and to the monolayer cell cultures, appropriate dilutions of compounds in the culture medium (200 µL/well) were added and incubated for an additional 24 h to assess the influence of different concentrations of compounds. The cell viability was determined by measuring the metabolic activity using a 3-(4,5-dimethylthiazol-2-yl)-2,5-diphenyltetrazolium bromide (MTT assay, Sigma Aldrich, St. Louis, MO, USA). After exposure to the compounds, the medium of each well was replaced with 10 µL of 5 mg/mL MTT stock solution diluted in 90 µL phosphate-buffered saline (PBS). After 2 h of incubation, isopropanol with 0.04 M HCl was added (100 µL/well). The absorbance was measured by a multiwell scanning spectrophotometer at 560 nm (Glomax, Promega, GmbH, Walldorf, Germany). The experiments were performed in triplicate.

#### 3.2.3. Clonogenic Assay

The cells were seeded in dilutions (1000 cells) on 6-well plates to assess the colony-forming properties after the therapy. Plates were placed in an incubator and left untouched for 10 days until colonies were observed in the control samples. After the incubation, DMEM was removed, and the cells were washed with PBS. Clones were stained with a 0.5% crystal violet mixture in 4% paraformaldehyde (PFA, Sigma-Aldrich, St. Louis, MO, USA) for 10 min. Afterward, the free stain was removed by washing with water and left to dry at room temperature. Next, only the eye-visible colonies (>~0.02 cm) were counted manually. The counting of the colonies was unbiased because the counting person was not familiar with the samples’ IDs. The experiments were performed in triplicate.

#### 3.2.4. Population Doubling Time

The population doubling time determines the dynamics of the cell culture development as the average time required for a cell to complete the cell cycle. In the case of cancer cells, population doubling time allows evaluation of the compounds’ efficiency. In the case of increased growth of the cell culture, the compound has a regenerative potential called cell self-renewal.

A total of 3 × 10^5^ A375, C32 and HaCaT cells were seeded in 35 mm culture dishes (Corning, New York, NY, USA). The cells were incubated at 37 °C in a humidified atmosphere containing 5% CO_2_. After 24 h, the culture supernatants were removed, and appropriate dilutions of compounds in the culture medium were added and incubated for an additional 24 h or 72 h. The cells were collected using trypsin and counted using KOVA (KOVA^®^ Glasstic Slide 10 with Grid Chamber, HYCOR Biomedical, Garden Grove, CA, USA) after 24 and 72 h.

#### 3.2.5. Cell Death Evaluation by Neutral Comet Assay

The neutral comet assay method was used to detect DNA damage associated with exposure to the used compounds [57,58]. Slides with cells were submerged in precooled lytic solution (100 mM EDTA, 2.5 M NaCl, 10 mM Tris base, 1% Triton X-100, pH 10) at 4 °C for 60 min. After lysis and rinsing, slides were equilibrated in TBE solution (40 mM Tris/boric acid, 2 mM EDTA, pH 8.3); after that, electrophorese was set at 1.2 V/cm for 15 min. To visualize comets, Sytox Green staining was performed (Thermo Fisher Scientific, Waltham, MA, USA) for the fluorescent microscope. For scoring the comet patterns, about 50 nuclei from each slide were assessed. CometScore 2.0 software was used to analyze the comets. The cell death type was assessed by the visual inspection described by Cortes-Gutierrez et al. (class 0—not affected, class 1 and 2—early apoptosis/intermediate damages, class 3—late apoptosis). The data are presented on the histograms [51].

#### 3.2.6. Fluorescent Staining of Actin Filaments

To visualize the actin filaments of the cells, confocal microscopy was used. The cells were incubated on cover glasses (24 × 24 mm, Thermo Fisher Scientific, Waltham, MA, USA) in 35 mm Petri dishes for 24 h with different concentrations (25–300 µM) of compounds. Control samples were prepared as well. Afterward, the cells were washed three times with PBS. Actin filaments were stained with Invitrogen™ Alexa Fluor™ 546 Phalloidin (2 μg/mL, A22283, Thermo Fisher Scientific, Waltham, MA, USA) with the manufacturer’s standard protocol. To stain cell nuclei, samples were fixed with a DAPI (4′,6-diamidino-2-phenylindole) solution (Roti^®^-Mount FluorCare DAPI, Carl Roth GmbH, Karlsruhe, Germany). The cells were examined using a Laser Scanning Confocal Microscope Olympus FluoView FV1000 (LSCM, Olympus, Warszawa, Poland). An oil immersion lens with 60x magnification, NA: 1.35 (Olympus, Tokyo, Japan) was used to capture the images.

### 3.3. Molecular Modeling—Computational Methodology

The structures of four receptors denoted as: cMet (PDB code: 3RHK) [52], EGFR (PDB code: 5GTY) [53], HER2 (PDB code: 7JXH) [54] and hTrkA (PDB code: 6PL2) [55] were used in the flexible docking study. They were taken from the Protein Data Bank (PDB) [59]. Hydrogen atoms were added with the usage of the Reduce program [60] and the “prepare_receptor” script (from the ADFR software suite) [61]. Subsequently, the AutoDockTools 1.5.7 program [62] was used to examine the macromolecule and the ligand to prepare them for further docking study. Every of the selected receptors was co-crystallized with the ligand in the binding pocket and the docking protocol was assessed based on the root-mean-square deviation (RMSD) values between the docked and the crystallized structures. The literature ligands have the following codes in the PDB database: M97 [52], 816 [53], VOY [54] and OOM [55], and were co-crystallized with the abovementioned receptors. Our flexible docking protocol included 150 independent searches with 3,500,000 evaluations of the scoring function. Flexible amino acid residues were chosen manually and, as a result, the binding pocket for each examined receptor had a different dimension. The dimensions and centers of the grid boxes of cMet, EGFR, HER2 and hTrka were set to: 27 × 1827 × 27; (−5.75, 12.46, −1.62), 32.2527 × 29.2527 × 28.50; (−35.75, 16.34, −61.24), 26.5027 × 25.7527 × 20.5; (65.75, 11.19, 82.58) and 20.5027 × 25.7527 × 34.00; (−18.27, −24.54, −18.85), respectively. The grid box sizes were adjusted to include all amino acids considered in the flexible docking procedure and were centered on their co-crystallized inhibitors present in the resulting pdb files. The binding affinity of the experimentally obtained ligands (compounds **3**–**12**) was compared to the drugs available on the market: Erlotinib (PDB code: AQ4) [63], Neratinib (PDB code: HKI) [64] and Tepotinib (PDB code: 3E8) [65]. The structures of the medicines were also downloaded from the PDB (they are further denoted as reference drugs). In the next step, the ligands (compounds **3**−**12** synthesized especially for the study) and the reference drugs underwent quantum-chemical simulations based on Density Functional Theory (DFT) [66,67]. The energy minimization was performed at the MN15/def2-TZVP level of theory using the continuum solvation model (IEF-PCM) with water as a solvent [68,69,70]. The harmonic frequencies were computed as well to confirm that the structures of the set of the studied compounds correspond with the minima on the Potential Energy Surface (PES)—no imaginary frequencies were found. For the set of compounds **3**–**12** and the reference drugs, the calculations of the Molecular Electrostatic Potential (MEP) were carried out using the DFT level of the theory mentioned above. The quantum-chemical simulations were performed with the Gaussian 16 Rev. C.0.1 suite of programs [71]. The Multiwfn and Visual Molecular Dynamics (VMD) 1.9.3 programs served for MEP calculations and visualization [72,73]. Subsequently, the “prepare_ligand” (from the ADFR software suite) script [61] was used to prepare ligands (in this case, the compounds denoted as **3**–**12** and the reference drugs) for the docking studies. The same docking protocol as in the case of the co-crystallized ligands was applied to estimate the binding affinities and the position of the ligand in the binding pocket of the selected receptors. The docking experiment and the ligand preparation were performed with the assistance of the AutoDock Flexible Receptor (ADFR) v1.2 suite of programs [74]. The 2D diagrams of the ligand–receptor interactions were generated in the BIOVIA Discovery Studio 2021 [75]. All visualizations of the binding pocket and the ligand positions were obtained with the ChimeraX 1.3 program [76].

## 4. Conclusions

In the present study, ten *N*-Mannich-base-type compounds **3**–**12** were synthesized for the first time, and their chemical structures were confirmed by detailed spectral analyses. All compounds were evaluated in vitro for their growth inhibitory activity on selected cancer cell lines: A375, C32, SNB-19, MCF-7/WT and MCF-7/DX. Two of the compounds (**5** and **6**) that displayed promising cytotoxic effect were further evaluated for anticancer activity on melanoma cells (A375 and C32) and human normal cells (keratinocytes) to explore their properties. The results of the MTT assay showed that the most significant cytotoxic effect was observed for compound **5** against A375 cells (IC_50_ = 80.79 µM). However, the highly cytotoxic impact of this compound on keratinocytes has to be considered as well. In the case of compound **6**, the anticancer effect was more selective to melanoma cell lines. A colony formation assay, population doubling time test and comet assay used in the neutral version, as well as fluorescent staining of actin filaments, proved the promising growth-inhibitory properties of compounds **5** and **6**. The results demonstrated the capability of the tested compounds to induce apoptosis and DNA damage in exposed melanoma cells; in particular, A375 cells were high sensitive to the genotoxic activity of compound **6**. Furthermore, this compound caused disturbing of the normal cytoskeleton organization by rearranging the F-actin microfilaments network in both melanoma cells at lower concentrations than those affecting normal keratinocytes.

However, the most promising compounds, from the perspective of the in silico flexible docking study to the selected receptors involved in cancer progression and metastasis (cMet, EGFR, HER2 and hTrkA) are compounds **7** and **11**. It is worth underlining that the binding affinities of the whole series of the synthesized compounds **3**–**12** are similar to or lower than the binding affinity of Erlotinib (one of the reference drugs). The MEP calculations revealed that compounds **3**–**12** are capable of interacting non-covalently. The interactions could be of the σ- and π-hole types.

Further research to investigate the mechanism of the anticancer effect of the tested compounds, including their implications in the cell cycle progression, as well as identify their molecular targets, are currently being investigated. Nevertheless, our present results constitute a foundation for further in vivo studies, which may lead to the selection of the most efficient compounds among the group of de novo synthesized *N*-Mannich bases of 1,3,4-oxadiazole based on a 4,6-dimethylpyridine core.

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
