# Peer review of "Synthesis, Anticancer Activity and Molecular Docking Studies of Novel N-Mannich Bases of 1,3,4-Oxadiazole Based on 4,6-Dimethylpyridine Scaffold"

_ijms, 2022, doi:10.3390/ijms231911173_

Round 1

Reviewer 1 Report

Present manuscript deals with the synthesis and anticancer activity of 2-mercapto-4,6-dimethylpyridine and N-piperazine derivatives linked via N-mannich bases of 1,3,4-oxadiazole and their molecular docking studies. The authors reported a total of ten new compounds of the series using N-piperazine substituted with aromatic and aliphatic moieties using Mannich reaction between the corresponding piperazine derivative and formaldehyde. These compounds were tested for their potential cytotoxicity against various cancer cell lines. Out of which compound 5 and 6 were where further evaluated in melanoma and normal (HaCaT) cell lines and claimed to have cytotoxic and proapoptotic effects. Although these newly designed derivatives are interesting and possess moderate cytotoxic activities, it needs further clear and detailed spectral characterization of this series of compounds. The overall manuscript needs major redrafting before it can be considered for re-review.

 1.       Introduction section needs to be redrafted. Authors should emphasize on literature as well as the outline of this study. The author should provide general information about their study and why the further extension of earlier reported compounds is important and what are the improvements current finding addresses.

22 Compounds synthesized in the manuscript need to be characterized correctly. Most importantly author must provide an elemental analysis of all the compounds. All 13C NMR spectrums are inconsistent with the structures and missing 2-5 carbon atoms in spectral characterization, this needs to be corrected or justified and provide all the spectrums (1H, 13C) in the supporting information.  

As claimed in Line 159-161: “The FT-IR spectra of compounds 3-12 showed peaks…….CONH functions.” does not distinguish between compound 2 and final compounds 3-12 as both the compounds possess amide bond.

4.       Authors need to modify the synthetic scheme 1 similar to the attached file. Also provide all the reaction and condition details in the scheme

5.       Add the following reference to the line 102-103: Oxadiazole rings are used as bioisosteres……………….improved metabolic stability”

DOI: 10.1016/j.bmc.2010.03.032 , https://doi.org/10.1039/C2MD20054F and DOI: 10.1021/jm2013248

6.       Most of the compounds reported in the manuscript showed moderate cell bioavailability except compound 5. Although compounds 3-8 have no substantial differences from the structural point of view, why does only compound 5 show better cytotoxicity?

7.       Author should provide all the biological experiments with appropriate positive control.

Author Response

Dear Reviewer,

We appreciate your time and efforts in reviewing our article. Thank you very much for all remarks and hints which helped us to improve our manuscript. All the issues indicated in the comments from Reviewer have been addressed.
We have clearly marked all changes to the manuscript text. We have also added required new tables, descriptions, explanations etc.
Please find below the answers to the Reviewer`s comments.

  1. Introduction section needs to be redrafted. Authors should emphasize on literature as well as the outline of this study. The author should provide general information about their study and why the further extension of earlier reported compounds is important and what are the improvements current finding addresses.

The introduction has been revised as suggested by the reviewer.

  1. Compounds synthesized in the manuscript need to be characterized correctly. Most importantly author must provide an elemental analysis of all the compounds. All 13C NMR spectrums are inconsistent with the structures and missing 2-5 carbon atoms in spectral characterization, this needs to be corrected or justified and provide all the spectrums (1H, 13C) in the supporting information.

NMR spectra were added to the supplementary materials. There are no results of the elemental analysis as the mass spectra of the high resolution spectrum have been taken. The absence of carbon signals in the 13C NMR spectra is characteristic of this type of spectra. This deficiency may be due to the symmetry of the molecule. In the 13C NMR spectra we observe the signals of not equal carbon atoms.

  1. As claimed in Line 159-161: “The FT-IR spectra of compounds 3-12 showed peaks…….CONH functions.” does not distinguish between compound 2 and final compounds 3-12 as both the compounds possess amide bond.

Substrates 1 and 2 have been published and characterized earlier. The information included in the description of the FTIR spectra was not intended to emphasize the differences between the substrates and products 3-12, but was intended to show the presence of an amide group.

  1. Authors need to modify the synthetic scheme 1 similar to the attached file. Also provide all the reaction and condition details in the scheme

Scheme 1 has been modified.

  1. Add the following reference to the line 102-103: Oxadiazole rings are used as bioisosteres……………….improved metabolic stability” DOI: 10.1016/j.bmc.2010.03.032 , https://doi.org/10.1039/C2MD20054F and DOI: 10.1021/jm2013248

References have been added to the manuscript

  1. Most of the compounds reported in the manuscript showed moderate cell bioavailability except compound 5. Although compounds 3-8 have no substantial differences from the structural point of view, why does only compound 5 show better cytotoxicity?

It is true that the compounds 3-8 are very similar structurally. However, when we analyze them from the perspective of the possible non-covalent interactions, then there is a possibility for compound 5 to take part in halogen bonding. Furthermore, the distinguishing feature between compounds 5 and 4 are their volume and the number of possible secondary bonds formed. To provide an evidence for that explanation we have also analyzed the ESPs for experimental hits (compounds 5 and 6). Relevant discussion was added in the main body of the manuscript, for clarity we attach it here, too:

For compound 6 only negative extrema are presented – in fact there is no σ-hole or π-hole at the CF3 substituent and at the center of the benzene ring, respectively. Due to that CF3 can act only as an electron density donor – what is a distinguishing feature between 5 and 6, because in the case of compound 5 we can observe two σ-holes (of magnitude 0.41 and 1 kcal/mol) on chlorine atoms attached to the benzene ring (in this manner compound 5 can form two halogen bonds).

  1. Author should provide all the biological experiments with appropriate positive control.

Thank you for the valuable comment. There was not planned to use any standard drug as a positive control. Probably, the comparison with the commonly used chemotherapeutic would be reasonable, but we have used ten tested newly synthesized compounds and various types of cancer cells. Despite this, there is no similarity with the most common cytostatic drugs, which were previously tested using MTT assay on various cell lines, also these used in our study e.g., doxorubicin (https://doi.org/10.1186/s12935-018-0625-9, DOI: 10.2147/IJN.S30445 ), cisplatin (doi: 10.12659/MSM.919786, doi: 10.18502/ijph.v50i5.6121 ) or 5-fluorouracil (doi: 10.1371/journal.pone.0056679, doi: 10.1016/j.ultrasmedbio.2006.01.011).

Reviewer 2 Report

The manuscript entitled ‘Synthesis, Anticancer Activity and Molecular Docking Studies of Novel N-Mannich Bases of 1,3,4-Oxadiazole Based on 4,6-Dimethylpyridine Scaffold’. The authors in their research article focus on the design, synthesis and anticancer evaluation of new series of N-Mannich base type hybrid compounds containing morfoline or different substituted piperazines moieties. Overall, the manuscript is well-written and presented. The results are adequately discussed and conclusions well-constructed. I have some minor points for authors. 

Page 3, line 132 - 134 – Authors could also indicate substituents types (R) of target compounds in Figure 1.

Page 5, line 174 - 177 – Why did the authors select these five human cancer cell lines for cytotoxicity evaluation?

Page 6, line 199 – Authors could incorporate standard error of the determination of IC50 for compounds 5 and 6 in Table 1. 

Page 7, line 200 – I suggest express the colony forming as ““log [CFU/mL]” instead “% of control cell” in Figures 4a and b.

Page 11, line 286 - 2911,3,4-oxadiazole analogs target some proteins/enzymes and these proposed by Authors may not be the “real” ones. Are there any published reports showing that cMet, EGFR, HER2 and hTrkA are cellular targets for 1,3,4-oxadiazole analogs?

Page 12, line 268 - 343 Based on docking studies, authors suggest that compound 7 can be potential “EGFR and HER2 inhibitor”, whereas for compound 8 can be potential “cMET and hTRkA inhibitor”. This needs to be demonstrated experimentally either in cellular system or using in vitro activity assays. In addition, potential compounds-binding regions were proposed in target proteins (Figure 10). However, key questions must be addressed: 1) Do the compounds-binding regions overlap with catalytic sites of target enzymes? 2) Are the amino acids involved in interactions with compounds important for enzymatic activity of target proteins? On the other hand, authors could perform calculation with other skin cancer-related proteins (such as kinase, oxidorreductase, oxygenase, etc.) in order to evaluate if experimental hits (compounds 5 and 6) have best in silico binding affinity than rest of compounds.

Page 14, line 348 - 351 Authors could analyze the MEP of the experimental hits (compounds 5 and 6) with those of reference drugs.

Page 19, line 621 - 623 – How did authors generate 3D-structure of the 3-12 compounds? Did authors use any software or website?

Page 20, Line 634 - 636 – More information about the docking procedure is needed, such as size and center of the grid box, and the preparation of the macromolecule. Was the grid box set for covering all the receptors or only the active site?

Author Response

Dear Reviewer,

We appreciate your time and efforts in reviewing our article. Thank you very much for all remarks and hints which helped us to improve our manuscript. All the issues indicated in the comments from Reviewer have been addressed.
We have clearly marked all changes to the manuscript text. We have also added required new tables, descriptions, explanations etc.
Please find below the answers to the Reviewer`s comments.

The manuscript entitled ‘Synthesis, Anticancer Activity and Molecular Docking Studies of Novel N-Mannich Bases of 1,3,4-Oxadiazole Based on 4,6-Dimethylpyridine Scaffold’. The authors in their research article focus on the design, synthesis and anticancer evaluation of new series of N-Mannich base type hybrid compounds containing morfoline or different substituted piperazines moieties. Overall, the manuscript is well-written and presented. The results are adequately discussed and conclusions well-constructed. I have some minor points for authors. 

Page 3, line 132- 134– Authors could also indicate substituents types (R) of target compounds in Figure 1.

Figure 1 has been improved. Additionally Scheme 1 shows the detailed structure of the new compounds.

Page 5, line 174- 177– Why did the authors select these five human cancer cell lines for cytotoxicity evaluation?

We wanted to test the activity on tumor cell lines, the treatment of which with currently available drugs is ineffective or impossible.

Page 6, line 199– Authors could incorporate standard error of the determination of IC50 for compounds 5 and 6 in Table 1. 

Thank you for the remark, we have corrected the Table 1 and included SE values.

Page 7, line 200– I suggest express the colony forming as ““log [CFU/mL]” instead “% of control cell” in Figures 4a and b.

Thank you for the suggestion. We have represented Fig. 4, where colony forming was expressed as “log [CFU/mL]”, however, we would like to save the previous representation, which more frequently used, and in this case more comparable in relation to the viability results obtained from MTT assay.

Page 11, line 286 - 291–1,3,4-oxadiazole analogs target some proteins/enzymes and these proposed by Authors may not be the “real” ones. Are there any published reports showing that cMet, EGFR, HER2 and hTrkA are cellular targets for 1,3,4-oxadiazole analogs?

We thank the Reviewer for the remark and question. We attach a set of publications related to the issue. The studies provided show that the 1,3,4-oxadiazole analogs are widely studied compounds for targets like cMet, EGFR, HER2 or hTrkA.

10.3390/molecules25215190

https://doi.org/10.1002/ardp.202200009

https://doi.org/10.3390/ph14090870

10.1016/j.bmcl.2008.07.057

https://doi.org/10.1080/15257770.2020.1761982

Page 12, line 268- 343– Based on docking studies, authors suggest that compound 7 can be potential “EGFR and HER2 inhibitor”, whereas for compound 8 can be potential “cMET and hTRkA inhibitor”. This needs to be demonstrated experimentally either in cellular system or using in vitro activity assays. In addition, potential compounds-binding regions were proposed in target proteins (Figure 10). However, key questions must be addressed: 1) Do the compounds-binding regions overlap with catalytic sites of target enzymes? 2) Are the amino acids involved in interactions with compounds important for enzymatic activity of target proteins? On the other hand, authors could perform calculation with other skin cancer-related proteins (such as kinase, oxidorreductase, oxygenase, etc.) in order to evaluate if experimental hits (compounds 5 and 6) have best in silico binding affinity than rest of compounds.

In-silico methods can serve us as some kind of screening, but it is not an evident argument against or for particular ligand or drug in terms of its bioactivity and efficiency. When it comes to the overlap of catalytic and compound-binding regions – to the great extent they overlap in our models. As it was mentioned in the computational methodology, the grid box was centered on already co-crystallized ligand of corresponding protein (cMet,EGFR,HER2 and hTrka), due to that most of the relevant amino-acids involved in the interactions with co-crystallized ligands are taken into consideration in our docking study. We attach papers relevant to all four receptors and their corresponding examined inhibitors:

cMet: https://doi.org/10.1074/jbc.M110.213801

EGFR: https://doi.org/10.18632/oncotarget.15443

HER2: https://doi.org/10.1158/0008-5472.CAN-21-2693

hTrka: https://doi.org/10.1016/j.bmcl.2019.126624

Page 14, line 348- 351– Authors could analyze the MEP of the experimental hits (compounds 5 and 6) with those of reference drugs.

We thank for the suggestion – we performed appropriate calculations and analyzed MEPs also of the abovementioned experimental hits (5 and 6). The data is presented in Figure 11. The new results were also commented in the manuscript body:

For compound 6 only negative extrema are presented – in fact there is no σ-hole or π-hole at the CF3 substituent and at the center of the benzene ring, respectively. Due to that CF3 can act only as an electron density donor – what is a distinguishing feature between 5 and 6, because in the case of compound 5 we can observe two σ-holes (of magnitude 0.41 and 1 kcal/mol) on chlorine atoms attached to the benzene ring (in this manner compound 5 can form halogen bonds).

Page 19, line 621- 623– How did authors generate 3D-structure of the 3-12 compounds? Did authors use any software or website?

The 3D structures of 3-12 compounds were generated with assistance of the Multiwfn program. We attach link to the website of Multiwfn code: http://sobereva.com/multiwfn/. We add a link to the original publication of Multiwfn program and its capabilities: https://doi.org/10.1002/jcc.22885.

In practice, to generate the structures one should obtain a .chk file of the converged molecular geometry and then to convert it to .fchk format. In the next step, the analysis with the Multiwfn program was employed.

Page 20,Line 634 - 636– More information about the docking procedure is needed, such as size and center of the grid box, and the preparation of the macromolecule. Was the grid box set for covering all the receptors or only the active site?

We provided additional information of the docking procedure in the Computational Methodology section in the manuscript. The added sentences are as follows:

Dimensions and centers of the grid boxes of cMet, EGFR, HER2 and hTrka were set to: 27x18x27; (-5.75,12.46,-1.62), 32.25x29.25x28.50; (-35.75,16.34,-61.24), 26.50x25.75x20.5; (65.75,11.19,82.58) and 20.50x25.75x34.00; (-18.27,-24.54,-18.85), respectively. The grid box sizes were adjusted to include all amino-acids considered in the flexible docking procedure and were centered on their co-crystallized inhibitors present in the resulting pdb files.

Preparation of the macromolecule:

We added a sentence describing that we used AutoDockTools 1.5.7 to prepare macromolecule and ligand for the docking study:

Hydrogen atoms were added with the usage of Reduce program [6] and the “prepare_receptor” script (from the ADFR software suite) [7]. Subsequently, the AutoDockTools 1.5.7 program [8] was used to examine the macromolecule and the ligand to prepare them for further docking study.

Reviewer 3 Report

The author should address the points given below.,

1. Explain the main consideration for the detection of different type cancer cells? Is there any specific reason to choose breast and melanoma cancer cells together? 

2. Which standard drug have been used for the MTT assay? Is there any one used? If not, explain the reason why nothing has been used?

3. Most of the time the IC50 values of the molecules are calculated after 72h after the incubation since the optimum duration for the drug effectiveness is obtained at that time? Why 24h has been taken for this assay?

4. Which solvent has been used for bio assay and what is the behavior of that solvent against the cancer and normal human cells? Also the author should give the final percentage of solvent in the bioassay?

5. In the results given for bioassay, 80 µM and the concentrations above this value can not be evaluate as potent molecules. The concentrations above 1 µM is usually toxic for normal human cells. Do the authors still think that the compounds are potent?

6. Another important issue that the author should explain is that, why the theoretical and practical results are not compatible? 

Author Response

Dear Reviewer,

We appreciate your time and efforts in reviewing our article. Thank you very much for all remarks and hints which helped us to improve our manuscript. All the issues indicated in the comments from Reviewer have been addressed.
We have clearly marked all changes to the manuscript text. We have also added required new tables, descriptions, explanations etc.
Please find below the answers to the Reviewer`s comments.

The author should address the points given below:

  1. Explain the main consideration for the detection of different type cancer cells? Is there any specific reason to choose breast and melanoma cancer cells together? 

Thank you for the comment. This study intended to deliver new data obtained from the evaluation of the newly developed compounds. Thus we have selected cell lines derived from various types of cancers. We hope this study will contribute to further research on these types of compounds and extend the knowledge of their mechanism of action in specific cells.

  1. Which standard drug have been used for the MTT assay? Is there any one used? If not, explain the reason why nothing has been used?

Thank you for the valuable comment. There was not planned to use any common drug cause no similarity with ten tested newly synthesized compounds. The most common cytostatic drugs were previously tested using MTT assay on various cell lines, also these used in our study e.g. doxorubicin (https://doi.org/10.1186/s12935-018-0625-9, DOI: 10.2147/IJN.S30445), cisplatin (doi: 10.12659/MSM.919786, doi: 10.18502/ijph.v50i5.6121 ) or 5-fluorouracil (doi: 10.1371/journal.pone.0056679, doi: 10.1016/j.ultrasmedbio.2006.01.011).

  1. Most of the time the IC50 values of the molecules are calculated after 72h after the incubation since the optimum duration for the drug effectiveness is obtained at that time? Why 24h has been taken for this assay?

Thank you for the remark. IC50 was calculated after 24h. Only the doubling time was measured after 24 and 72h. The rest of the study, including cytoskeleton analysis and apoptosis, was analyzed after 24h, because during this time, the most significant changes occur in cells. It is, in particular important in the case of early apoptotic changes.

  1. Which solvent has been used for bio assay and what is the behavior of that solvent against the cancer and normal human cells? Also the author should give the final percentage of solvent in the bioassay?

Thank you for this question. In the material and methods section was included that stock solutions were prepared in DMSO (dimethyl sulfoxide); and final compound dilutions were performed in Dulbecco’s Modified Eagle’s Medium, where DMSO concentration did not exceed more than 1% in the sample.

  1. In the results given for bioassay, 80 µM and the concentrations above this value can not be evaluate as potent molecules. The concentrations above 1 µM is usually toxic for normal human cells. Do the authors still think that the compounds are potent?

Thank you for the comment. We agree that 1 µM concentration of some anticancer commonly used chemotherapeutics. Here we have tested new compounds which active concentration was estimated in our research. This is the first study presenting the possible cellular effects of N-Mannich base-type hybrid compounds. We think that these compounds have anticancer potential in vitro also when using higher concentrations as 80 µM, that’s why we have also included normal cells for comparison. There is for sure a need to conduct further research on 3D cultures and in vivo. We also consider in our future plans to use alternative methods for drug delivery, including electroporation and encapsulation.

  1. Another important issue that the author should explain is that, why the theoretical and practical results are not compatible?

We thank Reviewer for the question. However, in order to answer it, we must take into consideration many factors. Firstly, in-silico methods, especially those based on molecular mechanics (developed taking into account the empirical findings and neglecting the electronic structure of the studied data) perform very well in the case of molecular systems, but are a rather rough approximation. Due to provided reasons, most of the force-fields used in the molecular docking are blind to the interactions that arise from the anisotropic charge distribution throughout the molecule – thus no interactions such as halogen, chalcogen, triel or tetrel bonds can be detected and taken into account. Another problem is related to the drug transport to the target and the drug stability in the living cell environment – from the perspective of the in-silico methods we cannot afford to answer questions related to these issues. Due to abovementioned reasons, the mode of binding, as well as binding affinity, can sometimes differ between the experiment and the docking study.